# In Vivo Pro-Inflammatory Effects of Silver Nanoparticles on the Colon Depend on Time and Route of Exposure

**DOI:** 10.3390/ijms25094879

**Published:** 2024-04-30

**Authors:** Wojciech Grodzicki, Katarzyna Dziendzikowska, Joanna Gromadzka-Ostrowska, Jacek Wilczak, Michał Oczkowski, Łukasz Kopiasz, Rafał Sapierzyński, Marcin Kruszewski, Agnieszka Grzelak

**Affiliations:** 1Department of Dietetics, Institute of Human Nutrition Sciences, Warsaw University of Life Sciences, 02-776 Warsaw, Poland; wojciech_grodzicki@sggw.edu.pl (W.G.); joanna_gromadzka-ostrowska@sggw.edu.pl (J.G.-O.); michal_oczkowski@sggw.edu.pl (M.O.); lukasz_kopiasz@sggw.edu.pl (Ł.K.); 2Department of Physiological Sciences, Institute of Veterinary Medicine, Warsaw University of Life Sciences, 02-776 Warsaw, Poland; jacek_wilczak@sggw.edu.pl; 3Department of Pathology and Veterinary Diagnostics, Institute of Veterinary Medicine, Warsaw University of Life Sciences, 02-776 Warsaw, Poland; rafal_sapierzynski@sggw.edu.pl; 4Department of Molecular Biology and Translational Research, Institute of Rural Health, 20-090 Lublin, Poland; m.kruszewski@ichtj.waw.pl; 5Centre for Radiobiology and Biological Dosimetry, Institute of Nuclear Chemistry and Technology, 03-195 Warsaw, Poland; 6Cytometry Lab, Department Oncobiology and Epigenetics, Faculty of Biology and Environmental Protection, University of Lodz, 90-236 Lodz, Poland

**Keywords:** silver nanoparticles, colon, intravenous, oral, oxidative stress, inflammation

## Abstract

Nanosilver is a popular nanomaterial, the potential influence of which on humans is of serious concern. Herein, we exposed male Wistar rats to two regimens: a repeated oral dose of 30 mg/kg bw silver nanoparticles (AgNPs) over 28 days and a single-dose injection of 5 mg/kg bw of AgNPs. At three different time points, we assessed antioxidant defense, oxidative stress and inflammatory parameters in the colon, as well as toxicity markers in the liver and plasma. Both experimental scenarios showed increased oxidative stress and inflammation in the colon. Oral administration seemed to be linked to increased reactive oxygen species generation and lipid peroxidation, while the effects induced by the intravenous exposure were probably mediated by silver ions released from the AgNPs. Repeated oral exposure had a more detrimental effect than the single-dose injection. In conclusion, both administration routes had a similar impact on the colon, although the underlying mechanisms are likely different.

## 1. Introduction

The last few decades have witnessed a rapid advancement of miniaturization, leading to the widespread usage of nanotechnology in different areas of science, industry and everyday life [1,2]. Useful physicochemical properties and efficient antibacterial action make silver nanoparticles (AgNPs) one of the most commonly used nanomaterials [2,3], often applied in diverse fields, such as the food industry, medicine, textiles, electronics, cosmetics or toys [1]. Due to their presence in commonly used products, AgNPs have penetrated into the human environment and should be treated as potentially hazardous pollutant [4]. In fact, evidence from both in vitro and in vivo studies coincide, in that nanosilver can exert a harmful influence on organisms [5]. Concomitantly, in animal models, AgNP accumulation, upon exposure via different routes, has been observed in many organs, from the liver, spleen, kidneys and lungs to the digestive tract and even the brain [4,6,7,8,9]. Once in the body, nanosilver can induce various toxic effects, such as cytotoxicity, genotoxicity, immunotoxicity or neurotoxicity [6,8,10,11,12,13,14]. The proposed underlying mechanisms involve oxidative stress, mitochondrial damage and inflammation, eventually leading to cell death. However, the exact mode of action of AgNPs is yet to be fully elucidated, since there are at least several factors that may affect their interaction with biological systems, such as particle size, shape or concentration, as well as the release of silver ions or surface functionalization [1,15].

AgNPs present in the environment can enter the organism via different routes, among which the most important are the oral and nasal routes, as well as intraperitoneal or intravenous injection [16]. The oral and nasal routes are relevant to environmental exposure, while intravenous and intraperitoneal administration may occur during clinical procedures. The former have been found to provoke adverse effects at the lowest doses and may originate easily from dietary supplements, food packaging or contaminated water consumption [4,17]. The latter, relevant primarily to medical applications, enable the direct entry of AgNPs into the systemic circulation, a process that could facilitate and accelerate their toxic action [17,18]. In order to compare these two important routes, we investigated the effects of intravenous and oral exposure to AgNPs on oxidative stress and inflammation-related parameters in the colon, plasma and liver of rats, following either a single-dose injection or a 28-day per os administration.

## 2. Results

### 2.1. Body Weight Gain

Animals were uniformly distributed in terms of body weight at the beginning of the experiment (no statistical differences between groups). Body weight steadily increased across the consecutive time points of the experiment, reaching a peak during week 4 of the treatment. Statistical analysis revealed that the weight changes did not depend upon the group, but only on the time (Figure 1).

### 2.2. Histological Evaluation of the Effect of Silver Nanoparticle Exposure on the Colon Tissue of Rats

Histological analysis of colon sections revealed a typical architecture of the colonic mucosa in all experimental groups. No significant change in the number of infiltrating lymphocytes was found between the control and AgNP intravenous (i.v.) group, 24 h after injection (Figure 2A,B). However, numerous small foci of lymphocytic infiltration in the intestinal mucosa were noticed in animals 28 days after AgNP i.v. administration (Figure 2D). In the control animals, a normal colon structure, without infiltrating lymphocytes, was observed (Figure 2C). No differences in the intensity of lymphocytic infiltration were observed between rats from the control group and the group treated with AgNP per os (Figure 3A–D).

### 2.3. Cholesterol and Liver Enzymes

Statistical analysis revealed that i.v. AgNP administration, unlike per os treatment, exerted a significant effect on cholesterol (CHO) plasma concentration, especially after 24 h, when the CHO level was higher in the AgNP i.v. group compared to the control. However, this difference might have occurred due to lower values of cholesterol concentration in relevant control animals. Activities of alanine aminotransferase (ALT) and aspartate aminotransferase (AST) were not affected by either of the experimental scenarios (Table 1).

### 2.4. Antioxidant Defense and Oxidative Stress Parameters

I.v. treatment with AgNPs disturbed the redox status of the colon (Figure 4). Statistical analysis demonstrated that the activity of glutathione peroxidase (GPx) was lower in the AgNP i.v. group, compared to the control, both after 24 h and 28 days (Figure 4A). It also significantly decreased during the post-treatment period. In contrast, AgNP-exposed animals showed elevated glutathione reductase (GR) activity 28 days after administration, in comparison with the untreated group, and the activity increased significantly throughout the post-experimental period (Figure 4B). Also, superoxide dismutase (SOD) activity significantly increased in animals exposed to AgNPs, compared to the control animals in both time points; however, no difference was observed between the two time points (Figure 4C). The reduced glutathione level significantly increased 28 days after i.v. treatment, compared to the relevant control group (Figure 4D). Additionally, it continued to significantly increase during the post-treatment period. In contrast, the oxidized glutathione level remained similar to the control at both time points but also significantly decreased during the post-treatment period (Figure 4E). Statistical analysis also showed an impact of the interaction between exposure and treatment duration on the glutathione (GSH) to glutathione disulfide (GSSG) ratio. It was significantly lower after 24 h and considerably higher on the 28th day in comparison with the control group (Figure 4F). At both time points, total antioxidant status (TAS) was lower in rats exposed to AgNPs than in the relevant controls, but the difference was not statistically significant (Figure 4G). No impact on lipid peroxidation products, in the form of thiobarbituric acid-reactive substances (TBARS), was detected (Figure 4H).

Significant alterations in rats’ intestinal antioxidant defense systems were also observed in animals administered with nanosilver per os. This effect was even more substantial than when administered intravenously. GPx activity significantly increased after 7 days of administration but returned to the control value after 28 days (Figure 4A). Both GR and SOD activities decreased after 7 days of administration but increased after 28 days, compared to the respective controls (Figure 4B,C, respectively).

The treatment also affected glutathione-related parameters. Similarly to GR and SOD activities, GSH concentration decreased after the 7-day administration, but increased after 28 days of exposure compared to the respective controls (Figure 4D). Conversely, GSSG concentration was highest after the first week of administration and reached the control level at the end of the experiment (Figure 4E). Consequently, the GSH to GSSG ratio followed the GSH pattern, decreasing during the 1st week of treatment and increasing at the end of experiment (Figure 4F). Statistical analysis also revealed a significant impact of the treatment on TAS and TBARS, which were higher in the AgNP-exposed animals after 7 and 28 days of treatment, compared to relevant controls (Figure 4G). However, the TBARS level significantly decreased in AgNP-treated animals between days 7 and 28. Nevertheless, it was still significantly higher than in control animals.

A comparison between the i.v. and oral administration showed that GSSG, TAS and TBARS values were significantly higher in the AgNP per os group compared to AgNP i.v. (Figure 4E,G,H, respectively).

### 2.5. Inflammatory Markers

The intravenous administration of nanosilver resulted in an inflammatory response in the digestive tract. AgNP exposure and its duration affected levels of interleukin 6 (IL-6) and interleukin 10 (IL-10) (Figure 5A,B, respectively). Statistical analysis revealed that the IL-6 level was significantly higher 24 h after i.v. exposure, whereas the IL-10 level increased considerably 28 days after exposure. The concentration of interleukin (IL-12) was significantly elevated at both time points, compared to the respective controls. However, similarly to the IL-6 levels, a significant decrease was observed when comparing AgNP-treated animals 24 h and 28 days after exposure. Nevertheless, the IL-12 level was still significantly higher than in control animals (Figure 5C). There were no differences observed in the concentration of tumor necrosis factor alpha (TNF-α) compared to the control (Figure 5D).

Oral exposure to AgNPs also led to an inflammatory response, as indicated by the results of statistical analysis. The treatment, its duration and the interaction between these two factors had a profound impact on the levels of pro- and anti-inflammatory cytokines in the colon. Specifically, levels of IL-6, IL-12 and TNF-α were significantly higher at both 7 and 28 days of treatment compared to the respective control groups. Furthermore, these levels increased significantly during the treatment (Figure 5A,C,D, respectively). Similarly, IL-10 levels exhibited a significant increase during the first week of administration and remained stable throughout the 28-day period. In addition, significant differences between exposure routes were observed on the 28th day of intervention. All inflammatory markers were consistently higher in orally treated animals compared to those treated intravenously.

## 3. Discussion

Oral ingestion is considered one of the primary routes for AgNPs to enter the human body. Despite this, there is a clear lack of human intake data, making any safety assessments challenging. Although an in vivo study involving a 14-day oral exposure of healthy volunteers to AgNPs from a dietary supplement showed no negative health consequences [19], results from animal experiments provide more specific insights into this topic. A comprehensive study on the subchronic oral toxicity of AgNPs in rats showed that the lowest observable adverse effect level (LOAEL) was 125 mg/kg bw, while the unobservable adverse effect level (NOAEL) was 30 mg/kg bw [20].

Nanosilver usage is becoming increasingly popular in healthcare settings, leading to the potential for direct contact of AgNPs with human blood. Nanoparticulate silver is incorporated into more than 140 dietary supplements and medical utensils. Some of those products have been identified as causing silver leakage directly to the bloodstream, such as central venous, hemodialysis and anesthetic catheters, novel drug-delivery systems and wound-healing products [21,22]. However, there is still limited knowledge about human intravenous exposure levels and the tolerability of AgNPs in a real-world bioavailability scenario. While data from animal studies allowed for the derivation of a provisional tolerable intravenous intake of AgNPs for humans at a level of 0.14 µg/kg bw/day, estimating the magnitude of realistic nanosilver intravenous exposure remains challenging, similar to the oral intake [23].

In this study we used both oral and intravenous administration protocols to compare both routes of exposure at the end of the 28-day experiment. The per os group received AgNPs at a dose of 30 mg/kg bw, which corresponds to NOAEL dose stated in previous literature [8], while the i.v. groups received AgNPs at a dose of 5 mg/kg bw based on our previous research [6,10,14]. Throughout the entire experimental period, no signs of acute toxicity were observed. Change in body weight, a marker of overall health status, showed a steady increase until the end of the experiment, with no discernible differences between the experimental groups. Similarly, no differences were seen after 28 days oral administration of 0.5 or 1 mg/kg bw/day of nanosilver [24]. Juling et al. [25] and Wen et al. [26] also found no body weight disturbances 24 h after a single-dose intravenous AgNP administration of 4 or 5 mg/kg bw. Nanosilver is known to induce hepatotoxic effects [9], typically expressed as changes in AST and ALT activity, as well as CHO in plasma [27]. Nevertheless, in this study, neither the injection nor oral administration of AgNPs induced significant changes in AST and ALT activity. These outcomes are in line with results of other investigations using similar dosages and exposure routes [8,20,26,28,29,30]. An increased cholesterol level is used in toxicological studies as a marker of liver function, which is one of the primary organs for Ag accumulation after intravenous AgNP administration, as shown in our previous study [6]. Intravenous administration of 5 mg/kg bw of AgNPs was reported to elevate the CHO level in blood plasma, although the effect failed to reach statistical significance [26]. Our study supported this observation, as the CHO plasma level in the AgNP i.v. group was significantly higher 24 h after exposure, compared to the respective control, but it returned to control levels by the end of the experimental period (28 days). There was no significant effect in regard to the AgNP per os group. The results of this study, which include no effects on body weight and liver function assessed based on AST and ALT activity, as well as CHO levels, suggest limited or no systemic toxicity or hepatotoxicity at the investigated doses and types of nanoparticles, aligning with the NOAEL value.

Oxidative stress is considered to be one of the major mechanisms of nanotoxicity induced by AgNPs in mammalian systems, including the digestive tract [15,31,32]. Hence, we examined several parameters, related to the redox balance in the colon, that encompassed the activity of key antioxidant enzymes, such as GPx, GR and SOD, which are integral components of the organism’s antioxidant defense system, working to neutralize reactive oxygen species (ROS) [33].

Intravenous exposure resulted in the mobilization of the antioxidant defense system. We observed an increased activity of SOD at both examined time points and of GR 28 days after injection in the AgNP i.v. group. In addition, this exposure had an impact on levels of glutathione. We noticed an increase in the GSH level 28 days after AgNP injection, and a decrease in the GSSG level across the experimental period. These observations suggest a gradual, compensatory response to the initial oxidative stress induced by nanosilver exposure that took place in the post-injection period, as both changes are significant if short-period and long-period responses are compared (Figure 4D,E). The observed increase in GSH level might have also occurred as a compensation of its depletion, due to binding to Ag+. Binding to the thiol groups of glutathione is a well-documented mechanism of detoxification of metal ions [34]. Furthermore, increasing GR activity over time may be partially responsible for the higher concentration of reduced glutathione observed 28 days after AgNP i.v. administration. Likewise, Wen et al. [35] reported a decrease in GSH levels in the liver after 7 days, following a single intravenous administration of 0.5, 2.5 or 12.5 mg/kg bw of AgNPs, corresponding with the initial depletion of GSH observed in this study. To the best of our knowledge, no data regarding the effect of i.v. administration of AgNPs on intestinal GR and GPx activity are available, but some indications could be drawn from other organs. In our previous study focused on the brain effects of i.v. nanosilver exposure [10], we found GR activity to be significantly elevated after a single-dose injection of 5 mg/kg bw of AgNPs, which is in line with the current study. Since the only negligible change in the GSSG concentration, TAS and TBARS was observed 24 h after AgNP administration, the effects are likely unrelated to H_2_O_2_ production. We propose that this phenomenon might be related to the mechanism of ion release from the AgNPs’ surface. Under physiological conditions, nanosilver is known to produce the so-called “Trojan horse effect”, which consists of the emission of Ag+ ions into the environment [15]. After intravenous administration, Ag ions, not nanoparticles themselves, are likely responsible for the observed effects [36]. The oral route of exposure led to more pronounced perturbations in the antioxidant system in the colon. The repeated 28-day administration of AgNPs lowered GR and SOD activities after 7 days but caused an increase in both parameters at the end of the intervention. In contrast, GPx initially increased significantly but returned to the level of the control group by the 28th day. These changes in GPx activity were accompanied by alterations in GSH levels. The administration of AgNPs for the first 7 days resulted in a decrease in GSH levels and an increase in GSSG levels, compared to untreated rats, which clearly indicated the stimulation antioxidant defense, confirmed by the stimulation of GPx activity. The presence of oxidative stress in the gut after oral administration was further confirmed by elevated TAS and TBARS levels. At the end of the experiment, the GSH levels remained elevated, while GSSG dropped to the same level as the untreated control group; these, together with the elevated GR activity, indicate a compensatory mechanism aimed at the restoration of low-molecular thiols homeostasis.

There are insufficient data regarding how oral AgNP exposure affects the activity of the intestinal antioxidative defense system. In a multi-dose experiment conducted by Gan et al. [30], which involved exposure to 10 and 50 mg/kg bw AgNPs coated with (or not) polyvinylpyrrolidone (PVP) administered orally for 28 days, differences in effects were observed, depending on the dosages and organs analyzed. In this study, PVP–AgNPs increased GSH levels in the liver and lungs, while pristine AgNPs reduced GSH but only in the lungs. SOD activity was higher in the liver with PVP–AgNPs and lower in the lungs with both types of nanosilver. Similar findings were described by Yousef et al. [37] in a long-term study on rats orally exposed to 50 mg/kg bw/day of AgNPs. The treatment caused a uniform increase in TBARS in the heart and lungs, as well as a significant decrease in the GSH level and the total antioxidant capacity. It had also an impact on SOD and GPx activities, both of which were considerably lowered in the exposed group.

Under physiological conditions, antioxidant enzymes interact with each other. Disturbance in the functioning of any of them yields a decline in the antioxidant defense. Modifications in the activity of glutathione-dependent enzymes and glutathione level indicate the pro-oxidative effect of AgNPs oral exposure. The increase in GSSG concentration after 7 days and the accompanying increase in GPx activity, supports this hypothesis. Studies have shown that the cellular toxicity of AgNPs arises from the generation of hydrogen peroxide [31], which can then be scavenged in the reaction of GPx, during which GSH is simultaneously oxidized to GSSG. In our study, the GSH to GSSG ratio increased across the intervention and was significantly higher in comparison with the respective control at the end of the experiment. This is a result of the increase in GSH concentration due to AgNP exposure. Exposure to various factors inducing oxidative stress leads to an increase in GSH concentration, which is considered an adaptive mechanism. We observed this effect in our study. The elevated concentration of GSH in the intestine may result also from its increased transport to this organ. The high activity of γ-glutamyl transpeptidase on the surface of intestinal cells facilitates the efficient uptake of GSH from the blood, which is crucial for maintaining the colon’s defense capabilities, which is the largest surface area of the human body that makes contact with the external environment. Maintaining an elevated GSH level over 28 days of exposure is also accompanied by an increase in GR activity, which is responsible for maintaining the appropriate level of GSH in the cell by reducing GSSG to GSH, which also may contribute to an increase in the level of GSH. This is in line with the TAS results, which are influenced by the level of GSH. The per os exposure also caused an increase in TAS at both time points, relative to the control group. Moreover, the TBARS level sharply increased after 7 days of administration and slightly decreased towards the end of the experiment, although it remained significantly elevated compared to the control. These outcomes suggest that oral treatment with AgNPs not only affected the antioxidant defense mechanisms of the colon, but also caused oxidative damage. The increase in TBARS is most often associated with an increase in ROS production. Elevated ROS generation leads to the formation of increased amounts of more reactive radicals and, as a result, lipid peroxidation. The influence of nanosilver was more pronounced at the beginning and lessened, although it remained notable, after 28 days of administration. These effects might be dose- and/or tissue-dependent, since Nakkala et al. [38] reported no changes in antioxidant parameters, such as SOD, GPx and GSH, in the serum of rats treated orally with lower doses of AgNPs for 28 days.

Our findings confirm that AgNPs induce oxidative stress and affect the antioxidant defense system. However, it is worth noting that the specific effects may vary depending on the route of exposure, as elucidated in the current investigation. An interesting aspect of our study is the comparison between the AgNPs administered intravenously and those administered orally on the 28th day of the intervention. We observed significantly elevated TAS, GSSG and TBARS levels in the orally treated group, while, in the i.v.-administered group, the effect was much less pronounced. This might be due to the different experimental design (single dose vs. continuous treatment); however, it might also confirm that different exposure routes may engender oxidative stress through distinct mechanistic pathways.

Another widely recognized mechanism of AgNP action in many organs, including the gastrointestinal tract, is the induction of inflammation [9,32]. In this study, we examined several inflammation-related biomarkers, including IL-6, IL-10, IL-12 and TNF-α [39,40,41,42]. We found that the intravenous administration of AgNPs had no effect on TNF-α; however, it did cause an increase in proinflammatory IL-6 and IL-12 levels shortly after the injection (24 h), which later decreased throughout the experimental period. Conversely, IL-10 showed a significant increase from 24 h after to the 28th day of the intervention, suggesting that the injection induced an inflammatory response, which later was steadily suppressed. Similar observations were described by Wen et al. [35], who exposed mice with induced non-alcoholic fatty liver disease to a single injection of either 0.5, 2.5 or 12.5 mg/kg bw of AgNPs. They found that a single injection of AgNPs increased liver inflammation, with elevated IL-6 and TNF-α levels after 7 days. On the other hand, De Jong et al. [43] reported that the prolonged intravenous administration of AgNPs (6 mg/kg bw) over 28 days reduced IL-6, IL-10 and TNF-α levels in exposed rats’ spleens.

As expected, oral treatment with AgNPs caused a notable inflammatory response. Levels of IL-6, IL-12, TNF-α and IL-10 increased across the experimental period and were elevated in both time points. Similar results were described by Sousa et al. [44] following 14 days of oral AgNP administration (1 or 10 mg/kg bw) to mice. They observed increased levels of TNF-α and IL-10 in the colon in both doses, as well as an increase in IL-6 and IL-12 treated with the lower, but not the higher dose. Our findings are also in line with those of Nakkala et al. [38], who also reported increased serum levels of proinflammatory IL-6 and TNF-α in rats exposed orally to 5 or 10 mg/kg bw (on alternate days) of AgNPs for 28 days. Also, after a longer administration period and a higher dosage of AgNPs, Yousef et al. [37] described similar effects and found IL-6 and TNF-α to be markedly elevated in the heart and lungs of exposed rats. Furthermore, a recent study by Ren et al. [45] focused on several metallic nanoparticles found that AgNPs in particular were vastly detrimental to the gastrointestinal tract of orally treated mice (5 mg/kg bw for 4 weeks). They observed the upregulation of several proinflammatory cytokines, including IL-6, IL-12 and TNF-α, a pattern similar to the one emerging from our experiment.

Overall, nanosilver induced inflammation via both exposure routes. However, in rats receiving AgNPs by injection, all of the parameters were significantly lower, suggesting that the repeated oral administration was much more detrimental to the colon than the single-dose intravenous exposure. Studies on the fate of AgNPs upon digestion reveal that, in the presence of proteins, they form larger clusters that eventually break down into de novo formed nanoparticles able to reach the intestinal wall [46,47]. In this context, our findings suggest that human exposure to nanosilver via oral intake could potentially result in detrimental health consequences, such as compromised intestinal barrier integrity, as demonstrated both in vitro and in vivo [48,49]. This, in turn, has been linked to the development of serious disorders, such as inflammatory bowel disease, non-alcoholic fatty liver disease, obesity, Parkinson’s disease and depression [50].

## 4. Materials and Methods

### 4.1. Nanoparticles Characterization

Silver nanoparticles of a spherical shape and with a nominal diameter of 20 ± 5 nm were purchased from PlasmaChem (Berlin, Germany). In order to prepare the particle solution, 5 mg of AgNPs were dispersed in physiological saline (800 µL). The suspension was then sonicated using a probe sonicator (Branson, Danbury, CT, USA) for 3 min with 420 J/m^3^ total ultrasound energy. Immediately after sonication, 10× concentrated phosphate-buffered saline (PBS) (100 µL) and 15% bovine serum albumin (BSA) (100 µL) were added to the solution. To avoid particle aggregation, suspensions were prepared directly before administration to animals. The detailed characteristics of the AgNPs used in the experiment were previously described by Lankoff et al. [51] (Table 2).

### 4.2. Animals and Experimental Design

The experiment was performed on 14-week-old male Fischer rats (strain: Fischer 344/DuCrl), purchased from Charles River, Sulzfeld, Germany. The animals were maintained individually in polyurethane cages, under standard environmental conditions (temp. of 23 °C, 60% relative humidity, 12 h light–dark cycle) and allowed ad libitum access to feed (Sniff^®^ Spezialitäten GmbH R/M-H, Soes, Germany) and water. The experiment was approved by the 3rd Local Ethical Committee in Warsaw and performed in compliance with 3R rules (Replacement, Reduction, and Refinement) and national law (Resolution # 35/2014).

Upon 10 days of acclimatization, animals (*n* = 60) were assigned to four groups: AgNP i.v. (*n* = 16), AgNP per os (*n* = 16), control i.v. (*n* = 13) and control per os (*n* = 15). The mean body weight of rats at the beginning of the experiment was 205.7 ± 9.2 g. Animals from experimental groups were administered with 20 nm AgNPs, either intravenously (5 mg/kg bw) as a single dose injected into the tail vein (AgNP i.v. group), or orally (30 mg/kg bw) by gavage for 28 consecutive days (AgNP per os group), in accordance with Organisation for Economic Cooperation and Development (OECD) 407 guidelines [52]. Rats allocated to control groups were treated with 0.9% NaCl solution, administered using the same exposure route as the corresponding AgNPs groups.

Animal behavior, growth rate, feed intake, activity and any clinical symptoms were monitored on a daily basis throughout the whole experiment. Animals from i.v. groups were euthanized on days 1 and 28 (AgNP i.v., control i.v.), while animals from oral groups were euthanized on days 7 and 28 (AgNP per os, control per os), both by cardiac puncture under isoflurane anesthesia (Baxter Healthcare, Warsaw, Poland). Blood was collected through heart puncture and put into ethylenediaminetetraacetic acid (EDTA)-coated tubes (K2 EDTA tubes, Profilab, Warsaw, Poland) and plasma samples were obtained by centrifugation at 3500 rpm for 15 min at 4 °C. Both whole blood and plasma samples were stored at −80 °C until corresponding biochemical tests were performed. Liver and colon samples were harvested, rinsed with ice-cold PBS and immediately frozen in liquid nitrogen and stored at −80 °C for subsequent analyses. The scheme of the experimental design is shown in Figure 6.

### 4.3. Histopathological Evaluation

Colon samples from each rat were fixed in 10% formalin and dehydrated using immersion in a gradient of ethanol solutions. The preparations were then rinsed with xylene, embedded in paraffin blocks and cut into 4 μm sections. Colon tissue sections were stained with hematoxylin and eosin (H&E) for evaluation under a light microscope (Motic BA400, Olympus Corporation, Tokyo, Japan) by a veterinary pathologist.

### 4.4. Cholesterol and Liver Enzymes

Activity of ALT and AST were measured using premade analytical kits (Human Gesellschaft für Biochemica und Diagnostica GmbH, Wiesbaden, Germany), based on kinetic methods recommended by the Expert Panel of the IFCC (International Federation of Clinical Chemistry, Milan, Italy). Plasma concentration of cholesterol (CHO) was determined colorimetrically using commercial kits (P.T.H. Hydrex, Warsaw, Poland).

Liver tissue was homogenized in PBS buffer (pH 7.4; 0.01 M phosphate buffer with 0.0027 M potassium chloride and 0.137 M sodium chloride) (Sigma–Aldrich, St. Louis, MO, USA) with 1 mM of EDTA in a volume ratio of 1:7 using a homogenizer (Bio-Gen PRO 200, PRO Scientific, Oxford, CT, USA). Homogenates were then centrifuged (Multifuge 3L-R, Kendro; 10,000× *g*, 15 min, 4 °C) and the supernatant was used in further analyses. The activity of ALT and AST in the liver was determined using analytical kits (HUMAN Gesellschaft für Biochemica und Diagnostica mbH, Wiesbaden, Germany).

### 4.5. Inflammatory Markers

Harvested colon samples were homogenized in PBS and subsequently analyzed. Concentrations of pro- and anti-inflammatory cytokines (IL-6, IL-12, IL-10) and tumor necrosis factor alpha (TNF-α) were determined by a competitive specific enzyme immunoassay (ELISA), following manufacturer’s guidelines (DRG Instruments GmbH, Marburg, Germany).

### 4.6. Colon Antioxidant Defense and Oxidative Stress Parameters

Fragments of colon tissue were homogenized using the Bio-Gen PRO200 homogenizer in 50 mM PBS (pH 7) with 1 mM EDTA (at a tissue to buffer ratio of 1:10). The obtained homogenates were centrifuged (10,000× *g*, 15 min, 4 °C) and supernatants were aliquoted for subsequent assays. Antioxidant enzymes, including GPx, GR and SOD activity was determined using Ransel, GR and Ransod kits (Randox Laboratories, London, UK), respectively. The total antioxidant status-TAS of the colon was measured using a total antioxidant status kit (Randox Laboratories, London, UK). Additionally, the concentration of the reduced-GSH and oxidized-GSSG forms of glutathione in the colon tissue was determined using high-performance liquid chromatography (HPLC), following the method described by Rebrin et al. [53] using a 4-channel electrochemical array for simultaneous detection. A mobile phase (pH 2.7), consisting of 25 mM monobasic sodium phosphate and 0.5 mM 1-octane sulfonic acid as ion-pairing agents and 2.5% acetonitrile, was used for the isocratic elution of sulfhydryls on a C18 column (5 M column, 4.6 × 250 mm, flow rate 1 mL/min)]. The pH of the mobile phase was adjusted with 85% phosphoric acid. With 2.5% acetonitrile in the mobile phase, retention times were 5 and 20 min for GSH and GSSG, respectively.

Level of lipid peroxidation was determined as a concentration of TBARS, i.e., malondialdehyde (MDA) and secondary products of lipid peroxidation [54]. Specifically, samples were mixed with 1% thiobarbituric acid (TBA) and 0.1 M H_2_SO_4_ and heated for 1 h at 90 °C, then cooled, mixed with n-butanol and kept in a freezer until the separation of two phases occurred. The upper phase was collected and incubated with TBA. Subsequently, samples were centrifuged at 2000× *g* for 5 min and their absorbance was measured spectrophotometrically at 534 nm wavelength. TBARS concentration, expressed as MDA equivalents, was calculated from a standard curve using 1,1,3,3-tetramethoxypropane, which yields MDA upon hydrolysis during the assay.

### 4.7. Statistical Analysis

Data obtained from the in vivo experiment and biochemical assays were analyzed using Statistica v. 13.3 PL software (StatSoft Polska Sp. z o.o., Kraków, Poland). The repeated measures analysis of variance (ANOVA) was applied to analyze body weight changes during the experiment. Cholesterol concentration and liver enzymes activity, inflammatory markers, antioxidant defense and oxidative stress parameters were analyzed using two-way ANOVA in groups receiving either injection or oral administration with their respective controls. The comparison between different exposure routes at the 28th day of treatment was conducted using Student’s *t*-test. Before each analysis the appropriate assumptions were verified: the equality of variance was tested using the Brown–Forsythe test and the normality of residuals distribution was tested with the Shapiro–Wilk test. Data that did not meet the assumptions were logarithmized. Tukey’s Honest Significant Difference (HSD) post hoc test was applied in order to verify differences between groups. The statistical significance of the results was set at *p* < 0.05.

## 5. Conclusions

In conclusion, our study investigated the effects of AgNP oral and intravenous administration. We observed no signs of acute toxicity throughout the experimental period. However, AgNP exposure had a significant impact on oxidative stress and inflammatory markers in the colon despite the administration route. Our results suggest that the mechanisms differ depending on the exposure route. After intravenous AgNP administration, oxidative stress induction probably is unrelated to H_2_O_2_ production. We speculate that this phenomenon is linked to the release of Ag+ ions from the surface of AgNPs, rather than the impact of nanoparticles themselves. On the other hand, after oral exposure, oxidative stress seems to be associated with increased ROS generation and lipid peroxidation. Furthermore, all inflammatory-related parameters were negatively affected by the oral exposure route. It suggests that AgNPs can produce systemic effects and affect organs beyond their direct exposure site. Overall, per os treatment appeared to be more toxic and detrimental to the colon of exposed animals.

## Figures and Tables

**Figure 1 ijms-25-04879-f001:**
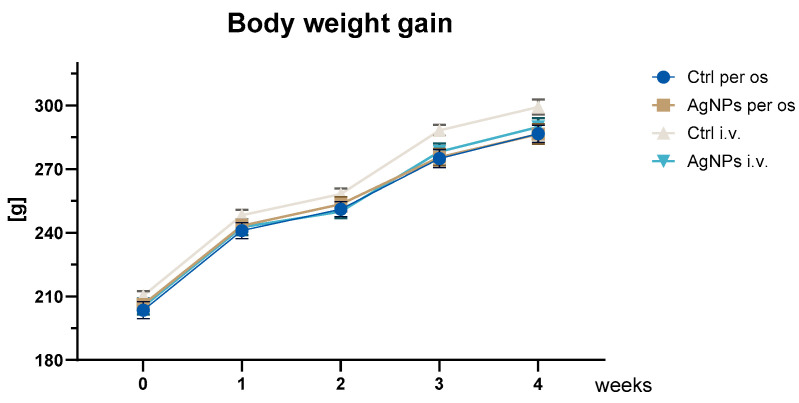
Body weight gain in treated and control (ctrl) groups during the experiment, mean ± SE; statistical analysis using two-way repeated measures analysis of variance (ANOVA) (time *p* = 0.001; time × group *p* = 0.016).

**Figure 2 ijms-25-04879-f002:**
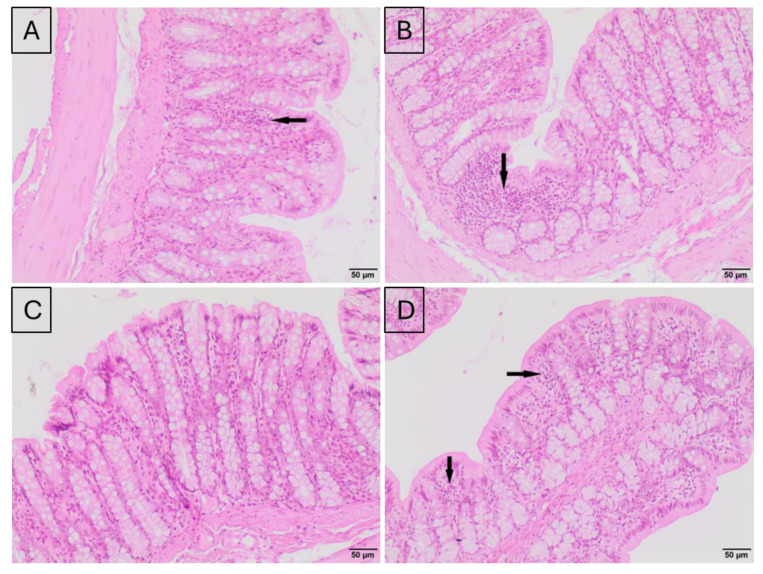
Representative microphotographs of hematoxylin–eosin-stained cross-sections of colons of rats from control groups: after 24 h (**A**) or after 28 days (**C**), and groups exposed, via i.v., to AgNPs: after 24 h (**B**) and after 28 days (**D**). (**A**)—a small focus of lymphocytes is visible in colonic mucosa; (**B**)—a small focus of lymphocytes is visible in colonic mucosa; (**C**)—normal structure of mucosa without lymphocytes; (**D**)—numerous small foci of lymphocytic infiltrate are visible in colonic mucosa. Hematoxylin–eosin staining; magnification 100×. Black arrows indicate accumulations of lymphocytes in the mucosa.

**Figure 3 ijms-25-04879-f003:**
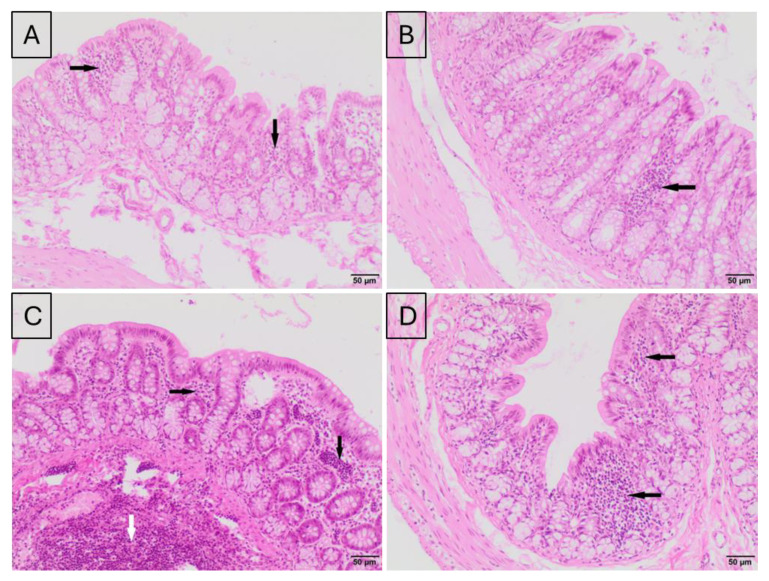
Representative microphotographs of hematoxylin–eosin-stained cross-sections of colon of rats relevant to per os treatment. Control groups: after 7 days (**A**) or after 28 days (**C**), and AgNP-treated groups: after 7 days (**B**) or after 28 days (**D**) of exposure. (**A**)—numerous small foci of lymphocytes are visible in colonic mucosa; (**B**)—one small focus of lymphocytes is visible in colonic mucosa; (**C**)—small foci of lymphocytes are visible in colonic mucosa, and accumulation of lymphocytes within muscularis. (**D**)—small numerous foci of lymphocytic infiltrate and one larger focus of lymphocytes are visible in colonic mucosa. Hematoxylin–eosin staining; magnification 100×. Black arrows indicate accumulations of lymphocytes in the mucosa, while the white arrow indicates accumulations of lymphocytes in the muscular layer of the intestinal wall.

**Figure 4 ijms-25-04879-f004:**
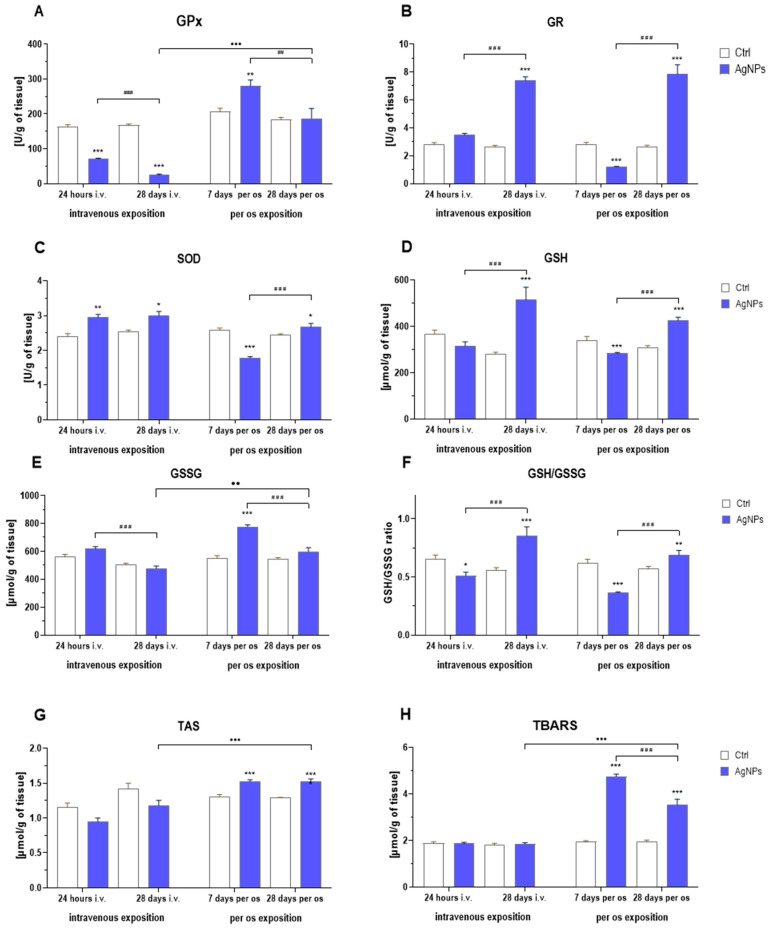
Antioxidant potential and oxidative stress in the colon of rats exposed to AgNPs via different routes (mean ± SE): (**A**)—glutathione peroxidase (GPx); (**B**) –glutathione reductase (GR); (**C**)—superoxide dismutase (SOD); (**D**)—glutathione (GSH); (**E**)—glutathione disulfide (GSSH); (**F**)—glutathione to glutathione disulfide concentration ratio (GSH/GSSG); (**G**)—total antioxidant status (TAS); (**H**)—thiobarbituric acid-reactive substances (TBARS). Data are expressed as mean ± SEM. *, ** and ***—significantly different from the respective control group (* *p* < 0.05, ** *p* < 0.01, *** *p* < 0.001) (Tukey’s post hoc test). ## and ###—significant differences between groups exposed to AgNPs in regard to time points (## *p* < 0.01, ### *p* < 0.001) (Tukey’s post hoc test). •• and •••—significant differences between groups exposed to AgNPs, with regard to the exposure route (•• *p* < 0.01, ••• *p* < 0.001) (Student’s *t*-test).

**Figure 5 ijms-25-04879-f005:**
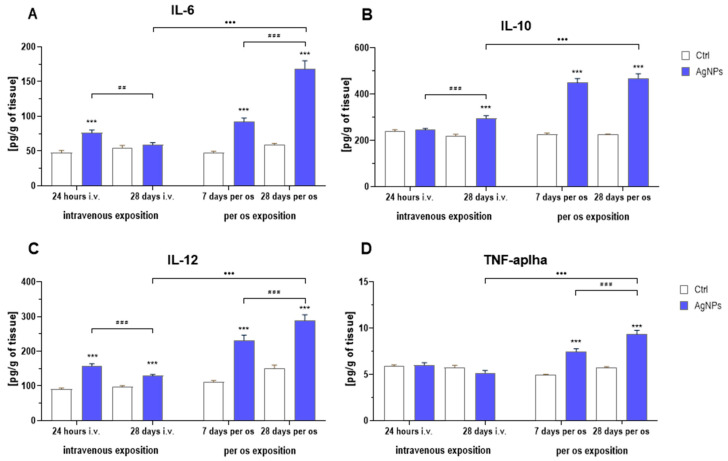
Inflammatory markers in the colons of rats exposed to AgNPs via different routes: (**A**)—interleukin 6 (IL-6); (**B**)—interleukin 10 (IL-10); (**C**)—interleukin 12 (IL-12); (**D**)—tumor necrosis factor alpha (TNF-α) (mean ± SEM). ***—significantly different from the respective control group (*** *p* < 0.001) (Tukey’s post hoc test). ## and ###—significant differences between groups exposed to AgNPs, with regard to time points (## *p* < 0.01, ### *p* < 0.001) (Tukey’s post hoc test). •••—significant differences between groups exposed to AgNPs, with regard to the exposure route (••• *p* < 0.001) (Student’s *t*-test).

**Figure 6 ijms-25-04879-f006:**
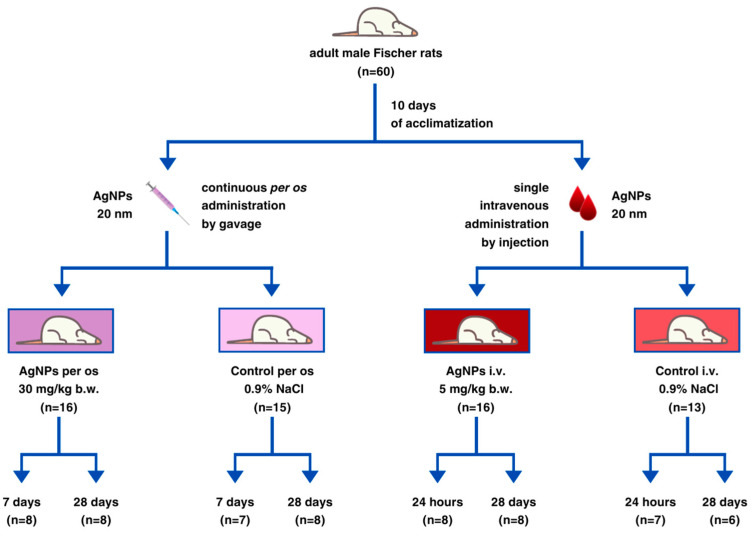
Scheme of the experimental design. Designed using elements by ©Canva via http://www.canva.com (accessed on 29 September 2023).

**Table 1 ijms-25-04879-t001:** Biochemical markers (mean ± SE).

	AgNPs				Ctrl			
	per os		i.v.		per os		i.v.	
	7 Days	28 Days	24 h	28 Days	7 Days	28 Days	24 h	28 Days
CHO [mg/dL]	50.2 ± 0.8	55.9 ± 1.7	55.3 ± 1.4 *	54.6 ± 0.8	56.6 ± 2.3	54.4 ± 2.1	48.7 ± 2.5	56.2 ± 1.3 ^a^
AST [U/L]	0.507 ± 0.024	0.482 ± 0.011	0.588 ± 0.024	0.553 ± 0.036	0.494 ± 0.028	0.532 ± 0.033	0.563 ± 0.045	0.498 ± 0.020
ALT [U/L]	0.831 ± 0.053	0.824 ± 0.030	0.870 ± 0.061	0.868 ± 0.045	0.881 ± 0.053	0.779 ± 0.021	0.823 ± 0.036	0.827 ± 0.034
AST liver [U/mg protein]	5.196 ± 0.506	4.822 ± 0.319	5.036 ± 0.615	5.585 ± 0.577	4.681 ± 0.375	4.230 ± 0.677	3.609 ± 0.124	5.510 ± 0.440
ALT liver [U/mg protein]	3.061 ± 0.253	2.846 ± 0.269	2.708 ± 0.242	3.163 ± 0.337	2.449 ± 0.186	2.307 ± 0.334	2.269 ± 0.217	2.828 ± 0.377

CHO—total cholesterol; AST—aspartate aminotransferase; ALT—alanine aminotransferase; ^a^—statistically significant difference from the other group, according to Tukey’s post hoc test (*p* < 0.05); the same letters show statistically significant results; * statistically significant difference from those of the control group, according to Tukey’s post hoc test (*p* < 0.05).

**Table 2 ijms-25-04879-t002:** Characteristics of AgNPs in water after dispersion, from Lankoff et al. [51].

	BSA-Coated AgNPs
Nominal size of Ag particles [nm]	20 ± 5
Dynamic light scattering [nm]	197.4 ± 2.7
Polydispersity index	0.295
Zeta potential [mV]	−33.6
BET surface area [m^2^/g]	2.2419
Micropore volume [cm^3^/g]	0.0076
Adsorption average pore width (nm)	13.6698
Desorption average pore width (nm)	23.8934

Data expressed as mean ± standard deviation (SD) (*n* = 3).

## Data Availability

The data that support the findings of this study are available on request from the corresponding author [K.D.].

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
