# Peer review of "In Vivo Pro-Inflammatory Effects of Silver Nanoparticles on the Colon Depend on Time and Route of Exposure"

_ijms, 2024, doi:10.3390/ijms25094879_

Round 1
Reviewer 1 Report
Comments and Suggestions for Authors
The article by Wojciech Grodzicki and colleagues provides in vivo pro-inflammatory effects of silver nanoparticles on the colon depends on time and route of exposure. The study mainly compares two routes of administration: oral exposure and intravenous injection. The authors addressed the main question of whether silver nanoparticles (AgNPs) can induce colon inflammation in rats by measuring oxidative stress and inflammation-related parameters and how the route of administration (oral vs intravenous) influences this effect. This provides insights into how the route of exposure influences the mechanisms involved. Thus the authors have addressed a gap by clarifying if route of administration plays a role in AgNP-induced colon inflammation.
The authors have made a significant contribution to the field by investigating the effects on oxidative stress and inflammatory markers (IL-6, IL-12, IL-10, TNF-α) in the colon.
The authors have used different methodologies to study the effect of AgNPs using histopathological evaluation, cholesterol and liver enzymes, inflammatory markers, colon antioxidant defense and oxidative stress parameters, etc.
The language is clear and professional throughout article but a few minor grammatical errors and awkward phrasings could be smoothed out. E.g., on line 355, 490.
The authors should use arrows to mark specific positions in the Figure 2 and 3. E.g. In Figure 2A, small focus of lymphocytes is visible in colonic mucosa. This will help the better readability of the figure.
In Figure 4 and 5, the authors should increase the font size on the X and Y axis.
The reference list is comprehensive and includes relevant literature. The authors need to include the most recent publications (2023, and 2024) in this study and discuss the results in the “Discussion” section regarding the effect of AgNPs induced colon inflammation.
Author Response
Dear Reviewer,
Thank you for your positive feedback on our manuscript. We appreciate your valuable suggestions, which will certainly help us improve our manuscript. The answers to the reviewers’ comments and proposed changes to the manuscript are enclosed below. The changes in the text have been highlighted in yellow. We hope that the manuscript is now suitable for publication in International Journal of Molecular Sciences.
Katarzyna Dziendzikowska, on behalf of the authors
- "The language is clear and professional throughout article but a few minor grammatical errors and awkward phrasings could be smoothed out. E.g., on line 355, 490."
The entire manuscript was carefully reviewed, and grammatical errors as well as awkward phrasings were corrected throughout, including lines 355 and 490. - "The authors should use arrows to mark specific positions in the Figure 2 and 3. E.g. In Figure 2A, small focus of lymphocytes is visible in colonic mucosa. This will help the better readability of the figure."
Arrows have been added to Figures 2 and 3 to mark specific positions as suggested. Additionally, we have included information about the arrows in the figure legends. Thank you for this helpful comment, which enhances the clarity and understanding of our results. - "In Figure 4 and 5, the authors should increase the font size on the X and Y axis."
We have improved Figures 4 and 5 by increasing the font size on the X and Y axes, as well as enlarging the parameter names to enhance readability. - "The reference list is comprehensive and includes relevant literature. The authors need to include the most recent publications (2023, and 2024) in this study and discuss the results in the “Discussion” section regarding the effect of AgNPs induced colon inflammation."
Thank you for your comment. We have carefully searched available databases and updated the literature list with new articles published in 2023. Additionally, we have incorporated two recent publications from 2023 into the discussion, both of which investigated the effects of orally administered AgNPs on rodents and reported similar outcomes, including an increase in pro-inflammatory cytokines. These references have been added as References 48 and 49. While we attempted to include publications from 2024, we found it challenging to find relevant studies that did not focus on fish, invertebrates, or in vitro models.
Reviewer 2 Report
Comments and Suggestions for Authors
The authors systematically investigated the effects of AgNPs on genotoxicity and neurotoxicity (Ref. 13, ref.14) as well as cytotoxicity in the colon, liver and plasma ( this study) of male rats. In this work, male Wistar rats were treated with AgNPs (20 nm) via oral administration and intravenous administration, respectively. They reported that AgNPs led to increased oxidative stress and inflammation in the colon as a result of either oral or intravenous administration. However, the induced toxicity mechanisms were different for these two routes.
The experimental techniques of this manuscript were planned carefully and carried out smoothly. In-vivo results were solidly explained and discussed. And the listed references were quite adequate. From these perspectives, I recommend the publication of this manuscript in present form.
Author Response
Dear Reviewer,
Thank you for your positive feedback on our manuscript. We are pleased to hear that our research on the impact of routes of nanosilver exposure on colon inflammation and oxidative stress in male Wistar rats met with your approval in terms of the experimental design, obtained results, and discussion.
Katarzyna Dziendzikowska on behalf of the authors